# Reliable Method to Detect Alloy Soldering Fractures under Accelerated Life Test

**M.A. Zamora-Antuñano** [1,†] , **O. Mendoza-Herbert** [2,†], **M. Culebro-Pérez** [2,†],
**A. Rodríguez-Morales** [1,†], **Juvenal Rodríguez-Reséndiz** [2,*,†] , **J.E.E. Gonzalez-Duran** [3,†] ,
**N. Mendez-Lozano** [1,†] and **C.A. Gonzalez-Gutierrez** [1,†]

1   Engineering Department, Universidad del Valle de México, 76230 Querétaro, Mexico
2   Facultad de Ingeniería, Universidad Autónoma de Querétaro, 76010 Querétaro, Mexico
3   Engineering Department, Instituto Tecnológico Superior del Sur de Guanajuato, 38980 Guanajuato, Mexico
*   Correspondence: juvenal@ieee.org
†   These authors contributed equally to this work.

**Abstract:**   In this research, we investigated the development and design of the Accelerated Life Test (ALT) and its approach to the waste of material. The development of a reliability model is based on the moment at which failure appears. The faults detected in welding joints during this research prevented proper current flow within electronic components and this interruption of current is considered a critical system failure. Minitab v18 was used to process data. Through statistical analysis, it was determined that the sample size was adequate with a 95% level of significance. A Shapiro Wilk analysis was carried out to determine the normality of the data, where a *p*-value of 0.1349 was obtained, which indicates that the data are normal. A Weibull analysis was applied, and it was observed that the data adjusted to the regression analysis and Weibull's reliability distribution. The results showed that failure phenomena can occur during electronic assembly due to the values of R being too high and too close to each other. Significant issues included the welding alloy, temperature, and the interaction between the welding alloy and vibration. It is observed that with high temperature, the number of faults in the solder alloy used for tin and lead and for tin, silver, and copper were lower. 17 electronic assemblies with measures of 2 cm × 2 cm were fabricated, where components such as leads and electric resistance were used. The objective of analyzing this is to obtain the characteristics of the soldering alloy. Electronic components of this type are used worldwide in all types of electronic components, including: TVs, cell phones, tablet, computers, resistors, diodes, LEDs, and capacitors. For this work, the components were built based on an LED and a diode.

**Keywords:** reliability model; time to failure; fracture; accelerated life test

## 1. Introduction

The Accelerated Life Test (ALT) is a method that can obtain data quickly to make decisions and improve efficiency and effectiveness in different types of products, to improve the manufacturing stages and raise the reliability of the conditions that may be higher than those originally designed. This work will allow researchers and interested companies to have information that addresses easy application methodology. The proposed method will allow more than just experts in the field to apply the ALT. This is because to apply different methodologies such as Quality Engineering, DMAIC, Design of Experiments, or others, it is required that people be trained for up to 200 man-hours, but the application presented in this work will allow use of this methodology easily and in a shorter time [1]. An electronic assembly is a system that carries out functions such as power control, communication protocols and telemetry systems in an industry. It could be used in a variety of elements such as a conveyor belt, a welder arm, or a packer. Most of the industrial processes have one or more electronic systems.

A soldering alloy is one of the most important elements of an electronic assembly as it carries the electric current that makes the assembly work, because the components, as part of the assembly, are joined to the motherboard, allowing the conduction of electrons. A failure in the soldering joint is a critical problem, because it will not allow current flow and the operation of the system will not work properly [2–5]. Some of the most common failures are open circuit, short circuit, cold weld, splash, weld bridges, porosity, and refluxes. All these problems can be caused by human error, random systems, or wasted material. Coit [4] indicated that an electronic assembly would fail if the soldering joints deteriorated to a critical level and decreased the reliability of the system. An open circuit could be caused by both thermal and mechanical stress, causing a ductile fracture in the soldering alloy. Mattila [3] indicated that a fracture appears due to impurities or phenomena originating during the welding process, e.g., the nucleation of micro-hollows or fractures for intergranularity. These impurities, scratches, or the remaining stress located on the surface of the weld cause the appearance of fractures. Therefore, it is important that the cleaning and finishing of the welding surface should be prepared in accordance with norms and codes. A ductile fracture occurs when the soldering alloy is exposed to plastic deformation due to stress caused by damage including excessive tensions such as thermal variation or mechanical effort. In some research [6–13], it is mentioned that mechanical effort causes parallel cracks, and thermal variation generates a network of multiple fissures. Furthermore, there have been fracture analyses using measures of resistance and visual inspection. Yang et al. [5] carried out a microstructural analysis of an electronic circuit that allowed visual identification of the propagation route of the fractures. In addition, they obtained micrographics using a high-resolution microscope. Certain authors have analyzed the different failures originating in soldering alloy. Parish et al. [6] observed that fluctuation in temperature caused expansions in the soldering alloy, generating additional stress and degradation. Due to this, there have been development tests that determine the capacity of a soldering alloy to resist thermal fatigue. Li et al. [7], Wu et al. [8], and Cheg [14] mentioned that electronic systems are a dynamic object susceptible to vibration. Their investigations demonstrated that a soldering alloy is a place of failure caused by vibrations. These authors used a criterion of failure as the identification of fracture propagation, which had a progressive increase of 20% of the initial resistance. The literature does not report many cases of tests with different combinations of tin that have been used in the development of this research: SnPb and SnAg are the most commonly used alloys in the market for the type of components presented in the present work. In the development of this research, 17 electronic assemblies with dimensions of 2 cm × 2 cm were fabricated, where components such as leads and electric resistors were used. This had the objective of analyzing the characteristics of the soldering alloy; see Figure 1. These types of electronic components are the most used worldwide in all types of electronics components, including TVs, cell phones, tablets, computers, resistors, diodes, LEDs, and capacitors.

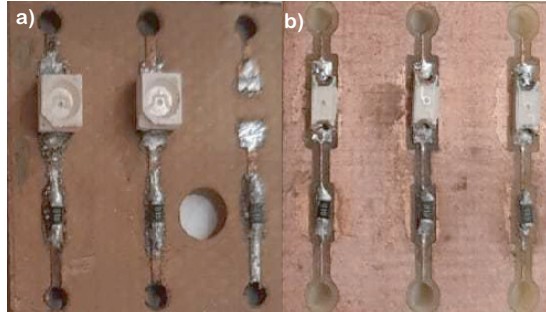

**Figure 1.** Electronic component assembly. Type Diode-LED. (**a**) shows SnPb Alloy. (**b**) shows SnAg Alloy.

*Objective*

In this work, we intend to detect faults through the application of the ALT in electronic components of SnPb and SnAg alloys using shorter analysis times.

## 2. Soldering Alloy

In the process of brazing, different metals such as tin (Sn), lead (Pb), antimony (Sb), copper (Cu), silver (Ag), cadmium (Cd), and zinc (Zn) are used. The metal welding for major applications in industry is lead, due to its low fusion point and price. It is cheap, and its fusion point is low. The disadvantage of lead is that it has a high grade of toxicity. However, the risk can be reduced if it is combined with other metals, thus obtaining alloys and reducing the toxicity. Another reason to obtain a soldering alloy is to improve the characteristics of the metals, such as its thermal and mechanical resistance. The soldering alloy mostly used with lead is tin, which has a different composition of tin with lead and a variation of the fusion-point alloy. The most common composition used in electronic applications is 60% tin and 40% lead, which has shown a decrease in the fusion point at 188 °C. Figure 2 shows an image of tin with lead obtained with an electron microscope. These alloys have a biphasic composition, i.e., the tin is observed at different temperatures to the lead. The lead-free solder is an alternative to reduce the toxicity of the lead; these types of alloys are more often rich in tin. The disadvantage of these alloys is that they have deficient wettability, and their fusion points are high, but their thermal resistance and their mechanical properties are higher. These types of alloys do not satisfy certain requirements of the 17 electronic components, forming ternary and quaternary alloys. Yang et al. [5], Parish et al. [6], Li et al. [7], and Wu et al. [8] mentioned that the most important tin ternary alloys are tin, silver, and copper (SnAgCu) and tin, silver, and bismuth (SnAgBi).

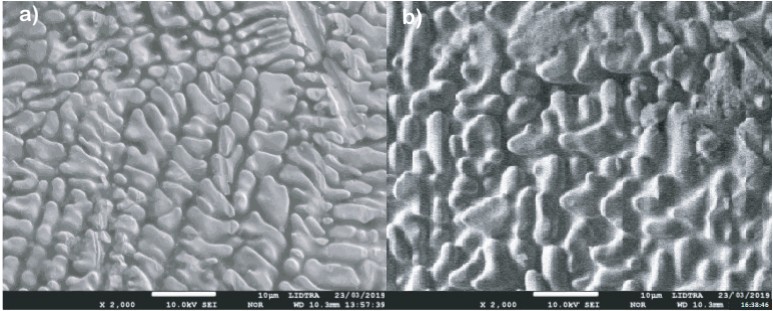

**Figure 2.** SEM image of biphasic composition of the tin and lead alloy taken at 2000×. (**a**) corresponds SnPb alloy. (**b**) corresponds SnAg alloy. (**a**) shows a dentric structure Sn base, secondary phase flat with microparticles precipitated in the dentric surface of the base of Pb. Spherical particles are observed.

## 3. Accelerated Life Test (ALT)

The ALT is used for determination of the reliability of the remaining lifespan of a component or a system. In this case, the soldering alloy test allows to identification of the exact moment when the soldering alloy fails. In the test performance, the accelerated methods are used, and these methods increase the experimental variable levels, accelerating the failure mechanism process as fatigue. Likewise, combinations of accelerated methods such as temperature and vibration could be used. In accordance with Wu et al. [8] and Kadanoff [9], the accelerated methods change the strain equivalent of the transformation of the time scale that will be used to register the failure time. The tests have two stages—physical straight models, and life distribution. In the first stage, the failure time is predicted. In a model with two variables, it is usual to consider the temperature and voltage or the temperature and humidity. The failure modes and the acceleration factors of the failure mechanism must be identified to choose these types of models. In the life distribution models, the exponential, Weibull, or Lognormal distributions are chosen. Likewise, many authors have been using analytical methods such as the Failure Mode and the Effects Analysis (FMEA) [9,10]. Furthermore, advanced

statistical methods as well as the Taguchi design have been used to select the best combination of parameters; however, the approach of variables includes all the electronic assembly [11–13,15–20]. In this research, a development and design of the ALT is presented, as well as its approach towards the material waste. Likewise, the development of the reliability model is based on the time when the failure appears.

## 4. Reliability Model

The objective in reliability research is to find failure time. Therefore, time is defined as *t* and is considered as a random variable. In this field, the reliability function indicating that the probability that the system will not present failures is defined as $Rh(t)$. According to Canakci [11] the reliability function and the failure rate are determined by the following equations:

$$R_h(t) = \int_0^\infty f_h(t)d(t) \tag{1}$$

where: $R_h(t)$ = Reliability function or survival function.

$$\lambda_h(t) = \frac{f_h(t)}{R_h(t)} \tag{2}$$

where:

- $\lambda_h(t)$ = Failure rate
- $f_h(t)$ = Instantaneous failure speed function

The mean time between failures (MTBF), according to Acevedo [12], is a function that measures the reliability of a reparable unit and is expressed in hours or years. This is determined with the following equation:

$$MTBF = \frac{TF}{n_{avg}} \tag{3}$$

where:

- $TF$ = Failure rate.
- $n_{avg}$ = Average of units examined.

The life of a system could be represented in the diagram of the Figure 3. This diagram is widely used to represent a remaining life curve where the 0 time to the $T_{n-1}$ time represents infantile mortal failures; the $T_{n-1}$ to $T_n$ represents random failures; the $T_n$ to $\infty$ represents wear failures.

As is known, the system failure causes are shown in a durability test, where the $\beta$ parameter must be calculated according to Acevedo [12], which could be interpreted as the following expressions:

- $\beta < 1$ is the risk rate decrease.
- $\beta = 1$ is the remaining life of a stable system, and the failures are random.
- $\beta > 1$ the risk rate increase indicating that the product is in the wear region.

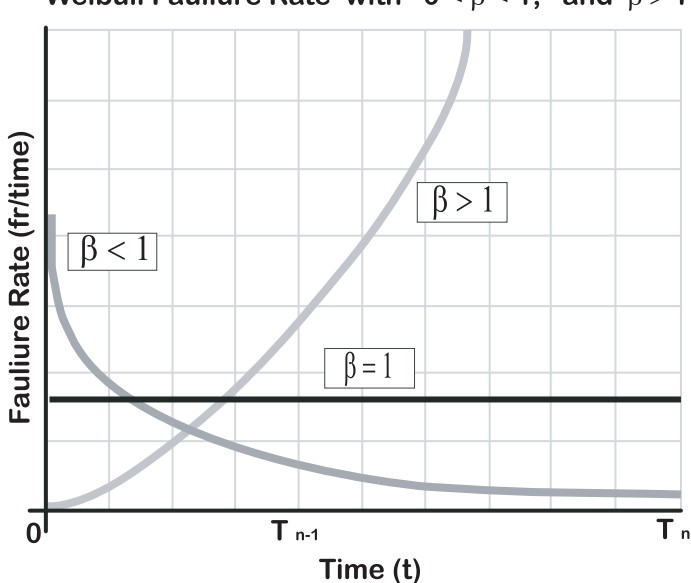

**Figure 3.** Weibull distribution. Adapted from the theories proposed by Ruiz et al. [10].

## 5. Methodology

As mentioned above, in the development of this research, 17 electronic assemblies with dimensions of 2 cm × 2 cm were fabricated, where components such as leads and electric resistance were used, with the objective of analyzing the characteristics of the soldering alloy. Electronic components of this type are among the most used worldwide in all types of electronics components such as: TVs, cell phones, tablets, computers; resistors, diodes, LEDs, and capacitors. For this work, the components were built based on an LED and a diode.

Five control factors were selected, as follows: soldering alloy, path, width, temperature levels, pretreatment, and vibration levels. The levels of the soldering alloy are Sn40Pb60 and Sn0.3Cu0.5Ag3.5, path 0.38 and 0.56 temperature 90 °C and 125 °C, pretreatment with thermal shock (0 °C) and without thermal shock, and vibration with resonance (40 Hz) and without resonance. The ALT was divided into two stages. In the first stage, the basis of the design of the test were established. In this stage, four tools were used, which are the functional block diagram, the Fault Tree Analysis (FTA), the FMEA, and the parameter diagram. To prepare these tools, field information, terms of use and technical sheets were used. At the other extreme, stage two consisted of the test design, where the failure modes, the stress variables, and the test conditions were established. In the failure mode effect analysis, the fracture in the soldering alloy was the highest failure mode with a Level of Risk Priority (LRP) of 300, which is due to the soldering alloy found in the pines of each component allowing the current flow. With the FTA, the causes of the failure were found, which are the temperature and vibration [9–13,15–20].

In the reliability test, it is important to know which of failure modes is in use, and what causes it. Due to this, analysis of the system must be done before proceeding with the tests. Therefore, information about the phenomena that originated the failure must be obtained, which will be the experimental variables. In this research, a design of experiment was used [20–22]. This allowed definition of the different combinations of the variables that duplicate the failure modes in minor time compared to the traditional methods. Table 1 shows the optimal experimental design obtained to realize the different combinations of the variables and its level, where -1 represents the low level and 1 represents the major level [10–13,15–23].

**Table 1.** Desing of Taguchi model.

| No. | Solder Alloy | Path | Temperature | Pret | Vibration |
|-----|-----|-----|-----|-----|-----|
| 3 | −1 | −1 | 1 | −1 | −1 |
| 4 | −1 | 1 | 1 | 1 | −1 |
| 5 | −1 | 1 | −1 | −1 | −1 |
| 8 | 1 | −1 | 1 | 1 | −1 |
| 11 | −1 | −1 | −1 | −1 | 1 |
| 12 | −1 | 1 | −1 | 1 | 1 |
| 13 | 1 | 1 | 1 | −1 | −1 |
| 15 | −1 | 1 | 1 | −1 | 1 |
| 17 | −1 | −1 | −1 | 1 | −1 |
| 18 | −1 | −1 | 1 | 1 | 1 |
| 19 | −1 | −1 | −1 | −1 | 1 |
| 26 | 1 | −1 | −1 | 1 | 1 |
| 28 | 1 | −1 | −1 | −1 | −1 |
| 31 | 1 | 1 | 1 | 1 | 1 |
| 33 | 1 | 1 | −1 | −1 | 1 |
| 39 | 1 | −1 | 1 | −1 | 1 |
| 45 | 1 | 1 | −1 | 1 | −1 |

Minitab v 18 [R] [24] made the corresponding calculations of the statistical analysis and the experiments design for calculating the accelerated factor of the temperature test. Arrhenius's equation was used (Equation (4)). This mathematical equation allows escalation of the time of the test with the time of use [12].

$$A_f = e^{\left[\frac{\ddot{E}_a}{k}\left(\frac{1}{T_1} - \frac{1}{T_2}\right)\right]}$$
(4)

where:

- $A_f$ = pre-exponential factor or frequency factor.
- $k$ = kinetic constant Boltzmann's constant (physical constant that relates absolute temperature and energy).
- $E_a$ = activation energy.
- $T_1$ = Field-use temperature
- $T_2$ = Accelerated test temperature.

Equation (5) exhibits the calculation of the acceleration factor of the temperature level 90 °C.

$$A_f = e^{\left[\frac{0.7}{0.000086}\left(\frac{1}{303.16} - \frac{1}{363.16}\right)\right]} = 84.42$$
(5)

The days of the year were divided by the acceleration factor, to calculate the equivalence on days of the test to a year in terms of use, as it is observed in Equation (6):

In electrostatics, the activation energy is known as the voltage threshold of diodes in general, and is equal to 0.7 V.

In the Arrhenius acceleration model and Ideal Gas Law, the Boltzmann's constant (k) has a value of $8.6 \times 10^{-5} = 0.0000086$

$$scale = \frac{365}{84.42} = 4.32$$
(6)

For the performance of the test vibration in a simulator, Shaker Tira Vib and a signal generator were used. This signal generator establishes the frequency and the width in which the Shaker will vibrate. Also, the current values were measured and the $\pm 2\,\mu A$ current was considered to be failure. To obtain specific data about the experimental unit, a vibration analyzer was used [25–30].

For the performance of the thermal test, a drying oven ECOSHELL, a j-type thermocouple and an acquisition data system was used. In the thermal shock, the samples were exposed to high and low temperatures. Measurements of current, temperature, and frequency were taken during the test.

## 6. Results

The results of the statistical analysis determined the power of the test that shows that the sample size is adequate with a significance level of 95%.

A Shapiro Wilk analysis was done to determine the normality of the data, where a *p*-value of 0.1349 was obtained, indicating that the data are normal. Tables 2–4 demonstrate the factorial regression obtained in the statistical analysis of design of experiments where the significant factors are the lineal model, the soldering alloy, the temperature, and the interaction with the soldering alloy and the vibration. The $R^2$ and $R^2$ adjustment are high and indicate that the model is useful for predicting the relationship between the time to failure and the significant factors. Table 4 shows the parameter of each significant factor to predict the time to failure [26,29,30].

**Table 2.** Factorial Regression: Failure Time vs. Soldering Alloy, Path, Temperature, Thermal Shock, Vibration.

|  | GL | SC Adjust. | MC Adjust. | $F_{value}$ | $P_{value}$ |
|---|---|---|---|---|---|
| Source | 5 | 726,392 | 145,278 | 27.39 | 0 |
| Model | 4 | 699,736 | 174,934 | 32.98 | 0 |
| Lineal | 1 | 531,227 | 531,227 | 100.15 | 0 |
| Soldering alloy | 1 | 21,415 | 21,415 | 4.04 | 0.07 |
| Temperature | 1 | 137,972 | 137,972 | 26.01 | 0 |
| Thermal Shock | 1 | 10,158 | 10,158 | 1.92 | 0.194 |
| Interactions with 2 terms | 1 | 24,321 | 24,321 | 4.59 | 0.055 |
| Soldering alloy*vibration | 1 | 24,321 | 24,321 | 4.59 | 0.055 |
| Error | 11 | 58,346 | 5304 |  |  |
| Lack of adjustment | 10 | 53,246 | 5325 | 1.04 | 0.649 |
| Pure error | 1 | 5101 | 5101 |  |  |
| Total | 16 | 784,738 |  |  |  |

The parameters of Table 2 are:

- *GL* = Degrees of freedom. Number of values that can be assigned arbitrarily, before the rest of the variables automatically take a value.
- *SC* = Sum of squares, represents a measure of variation or deviation with respect to the mean.
- *MC* = Square Medium, are used to determine if the terms of a model are significant.
- $P_{value}$ = probability corresponding to the test statistic.
- $F_{value}$ = is a value you get when you run an ANOVA test or a regression analysis to find out if the means between two populations are significantly different.
- *Adjust* = Adjusted Value, indicates which comparisons between the levels of the factors within a family of comparisons are significantly different.

**Table 3.** Summary of the model.

| S | $R^2$ | $R^2$-(Adjust) | $R^2$(Pred) |
|---|---|---|---|
| 72.8301 | 92.56% | 89.19% | 82.52% |

The parameters of Table 3 are:

- *S* = Standard deviation.
- $R^2$ = Coefficient of determination.
- $R^2(adjust)$ = Adjusted coefficient of determination. Is the percentage of variation in the response variable that is explained by its relation to one or predictor variables, adjusted for the number of predictors in the model.
- $R^2(pred)$ = Predetermined coefficient of determination. Adjustments in model looking for an approach in the desired values.

**Table 4.** Coefficients of the model.

| Term | Effect | Coef | SE Coef. | $T_{value}$ | $P_{value}$ | VIF |
|---|---|---|---|---|---|---|
| Constant | | 378.8 | 17.8 | 21.24 | 0 | |
| Soldering alloy | 361 | 180.5 | 18 | 10.01 | 0 | 1.04 |
| Path | −72.5 | −36.2 | 18 | −2.01 | 0.07 | 1.04 |
| Temperature | −184 | −92 | 18 | −5.1 | 0 | 1.04 |
| Thermal Shock | −51.5 | −25.7 | 18.6 | −1.38 | 0.194 | 1.1 |
| Solder alloy-vibration | −76.4 | −38.2 | 17.8 | −2.14 | 0.055 | 1.02 |

- $Effect$ = There is a main effect when different levels of a factor affect the response differently.
- $Coef$ = Coefficient of determination.
- $SECoef$ = The standard error of the coefficient estimates the variability between the coefficient estimates that would be obtained if the samples were taken from the same population repeatedly. The calculation assumes that the size of the sample and the coefficients to be estimated would remain the same if the sample were taken repeatedly.
- $T_{Value}$ = Measure the relationship between the coefficient and its standard error.
- $P_{Value}$ = probability corresponding to the test statistic.
- $VIF$ = The variance inflation factor indicates how much the variance of a coefficient is inflated due to the correlations among the predictors in the model. Use the VIF to describe how much multicollinearity (which is correlation between predictors) exists in a regression analysis. Multicollinearity is problematic because it can increase the variance of the regression coefficients, making it difficult to evaluate the individual impact that each of the correlated predictors has on the response.

Note: Table 4 shows the main factors to predict the failure time: Time to failure = 378.8 + 180.5 soldering alloy −36.2 path −92.0 temperature −25.7 thermal shock −38.2 soldering alloy-vibration.

Figure 4 shows the levels of each significant factor against the time to failure mean, where the level –1 has lower time to failure. The level 1 of the path has lower time to failure. Level 1 of the temperature has lower time to failure and the level –1 of the thermal shock has lower time to failure. All the levels of each factor have lower percentage of reliability.

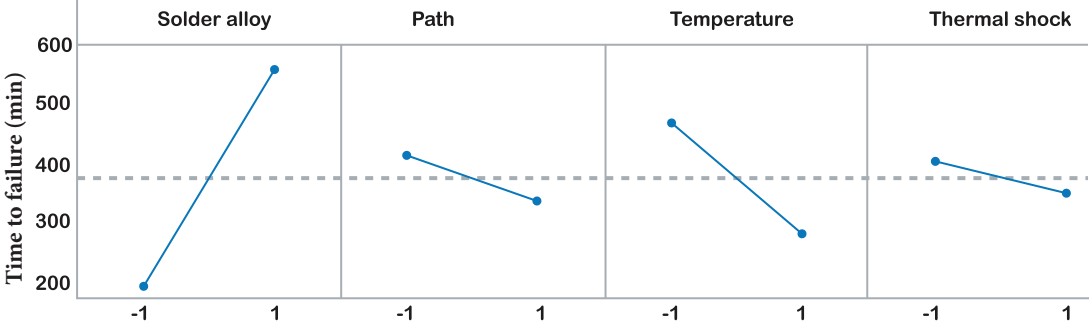

**Figure 4.** Principal effects graphic for times of fault.

Figure 5 shows an interaction graphic of the soldering alloy and the vibration levels where the interaction between the level +1 of the soldering alloy and the level –1 of the vibration has lower time to failure [25–31].

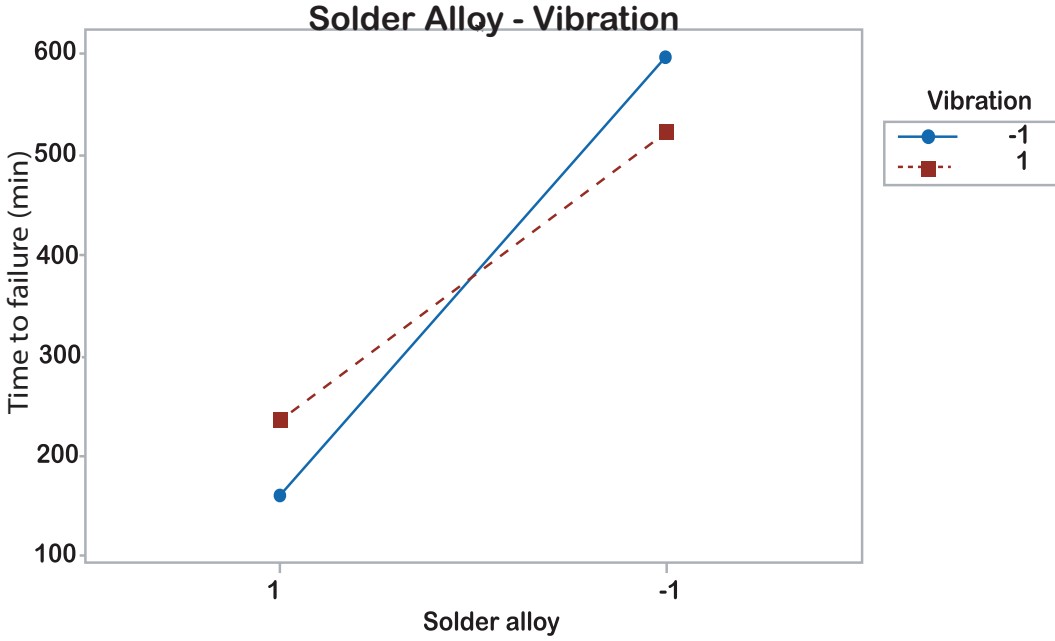

**Figure 5.** Interaction graphic for time to failure. Corresponds to the SnPB alloy.

The identification of failures in the welded joint was done by checking the continuity of the current with a multimeter [29–31]. However, the most important point in this research was the identification of fractures in the soldering alloy as critical failures in the welded joint. Images with an electron microscope were obtained to prove the aforementioned.

Figure 6 shows the microstructure of a SnPb solder alloy where a fracture could be observed in the material. Figure 6 shows a morphology of the fracture surface with SE (secondary electrons). Figure 6b is augmented 177×, and corresponds to the green box in Figure 6a. It consists of a typical ductile fracture of overlead. Particles of oxide are observed. The path of the fracture is through dendritic crystals. Figure 6d. Corresponds to the green box in Figure 6c shows the fracture zone where microcracks are derived. Figure 6d. A secondary crack of length of 8.14 μm, is observed.

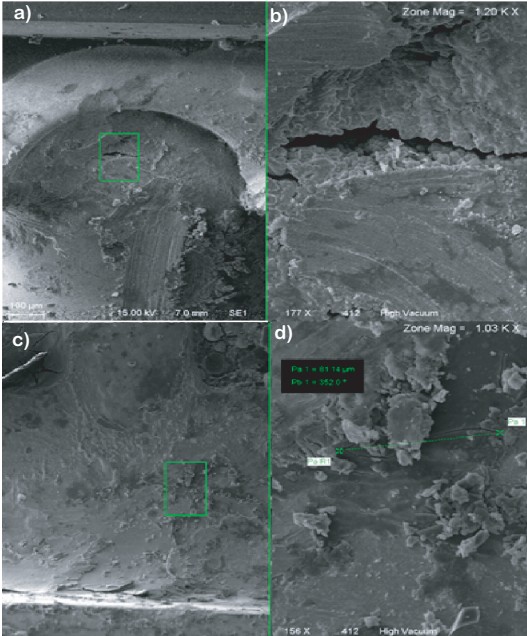

**Figure 6.** Fracture in the soldering alloy SnPb.

Figure 7a shows the fracture surface of one sample. A structure of primary solidification pattern consisting of a dendritic matrix of the SnAg alloy was observed. Figure 7b. The dendritic surface of the green box of Figure 7a. Shows larger dendrites Figure 7c. The green box shows the fracture zone of the SnAg alloy. Figure 7d. Corresponds to the green box of Figure 7c. It shows the surface of the crack where flat faces and oxides are observed.

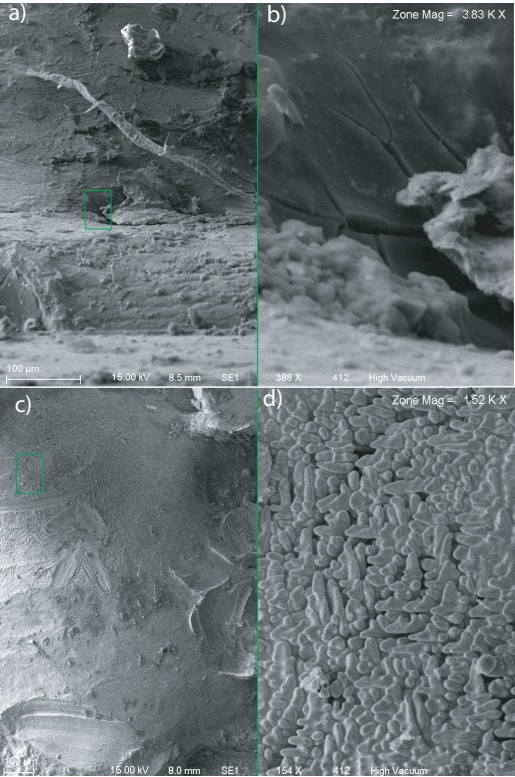

**Figure 7.** Fracture in the soldering alloy SnAg.

Figure 8 shows the unreliability graph of each level of the solder alloy, where the SnPb alloys showed lower reliability vs. the time. Table 5 shows the significant parameter of each soldering alloy where $\beta$ indicates failures due to the wear of the material.

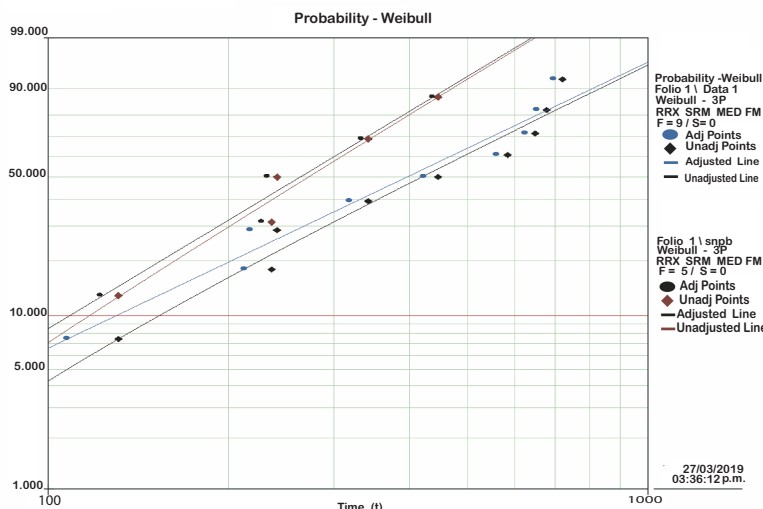

**Figure 8.** Unreliability graphic of the SnPb and the SnAgCu alloys. (The time is minutes).

**Table 5.** Reliability parameters.

| Soldering Alloy | Parameters |
|---|---|
| SnPb | Beta: 2.2923 |
| SnPb | Eta: 319.4363 |
| SnPb | Failure rate: 0.5364 |
| SnPb | Mean Life: 282.9822 |
| SnAgCu | Beta: 7.7790 |
| SnAgCu | Eta: 665.2968 |
| SnAgCu | Failure rate: $5.4508 \times 10^5$ |
| SnAgCu | Mean life: 625.6784 |

## 7. Discussion and Conclusions

### 7.1. Discussion

The results of the statistical analysis determined the power of the test that shows that the sample size is adequate with a significance level of 95%. A Shapiro Wilk analysis was done to determine the normality of the data, where a p-value of 0.1349 was obtained indicating that the data is normal. Also, the current values were measured and the $\pm 2$ µA current was considered to be a failure. Figure 5 shows the unreliability graph of each level of the solder alloy, where the SnPb alloys showed lower reliability against the time. In the Weibull distribution it is observed that the data did not form a curvature and that they are closed to the straight line; also, the values of $\beta$ greater than 1.0 indicated wear failures. With the results obtained, it is observed that over time the failure rate increases and the reliability decreases.

The main focus of the tests is to obtain data quickly, which when properly modeled and analyzed, provided the desired information about the life of a product under normal conditions of use. This process was developed to demonstrate that it is possible to find minor faults in Diode-LED electronic cards and resistors. There were no changes in the chemical composition as evidenced by micrographs. There was mechanical wear, cracks and ruptures. Due to the thermal shock of temperature and vibration, the porosity of the weld became larger due to the detachment of the weld. Other methods applied need more tests.

The method shows satisfactory performance at estimating the reliability of the system based on the reliability of the components. Moreover, using ALT provides estimates with lower bias and variance than using condition testing for this case.

In the literature review, the results obtained by different researchers can be checked. Thanks to these works, the results achieved could be validated and verified, and they allowed us to validate that the proposed method in their research works proposed provision of an effective solution for bearing fault prediction. The currently used high-lead-content solders (Snpb) are being applied for various applications irrespective of the operating temperatures (150–200 °C) and this is primarily because their chemical structure does not consist of any intermetallic compound. Thus, the operating temperatures do not have a major impact on their microstructure [32,33]. The aspect could have had an impact on the results achieved since the solder alloy of SnPb lead could fail due to the quality of the chemical manufacturing composition.

ALT consists of putting into operation a sample of some products of interest during a certain time under a controlled environment to obtain quick information about lifespan. The units under test are subjected to high-stress levels and they fail earlier than under design conditions. The information obtained under accelerated conditions is analyzed in terms of a model and then extrapolated under design conditions to estimate the life distribution and probability of failure. Therefore, metals, plastics, batteries, adhesives, semiconductors, insulators, bearings, chemicals, paints, asphalt, solar radiation, medications, single shot devices, maintenance planning, and even life situations may require degradation approaches. The theory has been described by Meekerand and Nelson [34] and Bai, Chung, and Chun [35] and these sources have used simple examples fit the theory.

When sample sizes are small, and the measurements show variability, the simple theory may be difficult to apply. For a single voltage at a constant level, being able to measure degradation in the presence of "noise" can be a challenge. A good foundation in Fault Physics (POF) can help solve the modes and causes of failure while monitoring degradation. The documents have limits for the application of accelerated degradation tests and at least seven assumptions for the successful use of degradation methods. A broader list includes:

- The observed degradation (change) is not reversible when tension is removed.
- There is a unique or dominant degradation process that can be studied (POF).
- Any degradation before the start of the accelerated test.

When accelerated life tests are used in industrial contexts, the efficiency of processes or assets is a problem for both manufacturers and end users. The reliability of the systems is a fundamental issue. Reliability assessment is usually based on the collection of field data during daily work. A common problem would be the sensing of data and its compilation since it is an activity that in many cases implies time and this due to the periods required for against with data that prove reliable.

It is important to mention that the ALT is a method to evaluate the components and the reliability of the systems in a short period, usually weeks or months, using a test strategy with excessive effort. In this work, a case study on welding in electronic components is discussed. The model for ALT studied here adjusts appropriately to the data of the case allowing a good estimate of the probability of failure, as shown by the tests performed for such adjustment to the level of confidence shown, although in general, these can change depending on the level that is required, but they would not do it radically. A review of the ALT that are necessary to model the lifetime of the units that underwent the accelerated test was made. The optimal plans for the mathematical model were studied with different considerations. Initially, it was assumed that the test would end until component failures occurred, but the case in which the test has a predetermined duration was considered. An optimal plan for an accelerated test with stepped efforts is developed for the Weibull model. For our case, the optimum time of application of the probability of failure under this test it was obtained, and that the test has a predetermined duration T. From the ALT test, the necessary information was obtained to determine the optimal plan for an accelerated test with two staggered efforts, following the procedure developed in this work. For our case we obtained the optimal time of application of the probability of failure under this test, and that the test has a duration T, predetermined. A set of experimental tests under stress conditions was developed and managed in 17 samples.

The results of the ALT modeling were compared with a Weibull Modeling based on real data and analyzed. In fact, the ALT model and the real data model are in agreement: the difference between the two average times and the failure estimates, was a minimum percentage, the quantitative tests of accelerated life, unlike the qualitative test methods, consist of tests designed to quantify the life characteristics of the product, component or system under normal conditions. Conditions of use, and therefore provide "information reliability". Reliability information may include:

- The determination of the probability of failure, the product under conditions of use, means low life.
- Conditions of use and projected returns and guarantees.
- Costs can also be used to help in the detection of faults and process improvement
- Carrying out risk assessments, design, comparisons, etc.

ALT can take the form of "Acceleration of the rate of use" or "Overload and acceleration". The ALT methods due to the acceleration of the usage rate and the test data can be analyzed with typical life data and methods of analysis, the acceleration of excessive stresses [32,34–38].

In the modern industrial context, the efficiency of assets is a big problem for both manufacturers and end users. The purpose of the known components and the reliability of the systems is a fundamental issue. The reliability assessment is normally based on the collection of field data during the daily work of the assets [34–37]. If the data collection is not automated, it is a laborious activity due to a very long period required.

The failure of the product is the event we want to understand. In other words, if we want to understand, measure and predict any event, for products that do not work continuously under normal conditions, if the test units run continuously, failures are detected earlier if the units were tested in normal use. Situations arise in life tests where the first failures are not reported, for example. a technician believes that early failure is "your fault" or "premature" and should not be recorded. Consequently, the reported data come from a truncated distribution and the number of uninformed premature failures is unknown. Inferences are developed for a Weibull ALT model in which transformed scale and shape parameters are expressed as linear combinations of environmental functions (stress) [11–13,15–18,32,36–38].

The coefficients of these combinations are estimated by maximum likelihood methods that allow calculating the point, interval and confidence limit estimates for said quantities of interest for a given voltage level, such as the shape parameter, the scale parameter, a quartile selected, the reliability at a particular time, and the number of early failures not reported. The methodology allows that the lifetimes are reported as accurate, censored to the right or with interval values, and are optionally subject to test protocols that establish thresholds below which lifetimes are not reported. A broad spectrum of applicability is anticipated by virtue of the substantial generality accommodated both in the stress model and in the type of data. Extensive material has been published in terms of ALT procedures and industrial experience over the years, creating technical knowledge in the execution of ALT adapted to the needs of each entity [14,33,39–41].

However, a new impulse in ALT now requires development, the creation of ALT procedures that can be more detailed. Life estimates applicable to new developments and new technologies as a result of this investigation, identified some areas as potential for further development:

- Application of Planned Statistical Techniques to expand the designer's ability to draw conclusions from an ALT;
- Study of strengths and weaknesses in each ALT model presented and cross this study with current practices to determine the best test strategies;
- Reliability management practices, with definition of sequence, importance, allocation of resources and impact of ALT procedures in the product development cycle.

### 7.2. Conclusions

It is observed that the data adjusted to the regression analysis and the Weibull Reliability distribution. The data indicated that they can represent the failure phenomena in electronic assembly, because the R values are high and close. In addition, with these results we can observe that significant factors are the soldering alloy, temperature, and the interaction between the soldering alloy and the vibration. It is observed that with a high temperature the number of failures in the soldering alloy of both tin and lead, and of tin, silver, and copper were minor. It is important to mention that the two temperature levels present overlap in their ranges of responses, so the difference between both levels is not significant [16,17]. It can be concluded that the temperature is important to the reliability of the soldering alloy but it is necessary to explore with other values to observe if there could be a major difference between levels [29–31,34,42].

It is concluded that the soldering alloy of tin and lead could fail due to the quality faults of the chemistry manufacturing. The soldering alloy of tin and lead used was a soldering paste which already has a flux. The flux is used to facilitate a uniform distribution of the tin in the joint avoiding an oxidation. When soldering paste is used, it is not necessary to use a flux and because the process of distribution is unknown. Due to this, future research is suggested using the soldering alloy of tin and lead and studying the type of supplier. The flux could be a study factor in the reliability of the soldering alloy. In the Weibull distribution it is observed that the data did not form a curvature and that they are close to a straight line; also, the values of $\beta$ higher to 1.0 indicated wear failures. With the obtained results, it is observed that over time the failure rate increases and the reliability decreases. It

is concluded that the design of the experiment in the development of the ALT could be used where the time of the test was shorter and therefore the results could be found more rapidly.

Through the method proposed in the present work, the objective was fulfilled, since the times to detect faults in components made of SnPb and SnAg alloys were reduced.

## 8. Concluding Remarks

In future research, tin and lead solder alloy should be used, alongside studying the type of supplier. It is important that in future research, flow can be a factor in the reliability of the welding alloy. Each supplier manages different welding manufacturing processes, with SnPb and SnAg, generating different results by dint of not following the same manufacturing process.

**Author Contributions:** Conceptualization, M.A.Z.-A. and J.R.-R.; methodology, M.C.-P.; software, A.R.-M.; validation, O.M.-H., J.E.E.G.-D., and J.R.-R.; formal analysis, N.M.-L.; investigation and visualization, M.C.-P., and J.R.-R.; data curation, J.E.E.G.-D., C.A.G.-G., and J.R.-R.; writing—original draft preparation; writing—original draft, review, and editing, all the authors.

**Funding:** This research was not funded.

**Acknowledgments:** The authors would like to thank the National Research Vice Rectory of the Universidad del Valle de México for their contribution of resources and materials to this research. The authors thank the assistance provided by Universidad Autónoma de Querétaro. The authors thank especially to the Master María Culebro Pérez, who has been the Project Leader. PhD Mario Enrique Rodríguez García and Eng. Carlos Ramírez Baltazar for the technical review and support in microscopy. In particular to the Master Aaron Rodriguez Morales, for the technical review and for his advice in the English Language.

**Conflicts of Interest:** The authors declare no conflict of interest.

## Glossary

| | |
|---|---|
| ALT | Accelerated Life Test. |
| MTBF | Mean time between failure. |
| TF | Failure rate. |
| $A_f$ | Pre-exponential factor or frequency factor. |
| $E_a$ | Activation energy. |
| $f_h(t)$ | Instantaneous failure speed function |
| $l_h(t)$ | Failure rate |
| $R_h(t)$ | Reliability function or survival function. |
| $\lambda_h t$ | Failure rate. |
| k | Boltzmann's constant (physical constant that relates absolute temperature and energy. Commonly for an LED 0.7 V is required). |
| $T_1$ | Field-use temperature |
| $T_2$ | Accelerated test temperature. |
| $\beta$ | Weibull distribution form parameter |
| $\eta$ | Scale parameter or Weibull distribution life parameter |

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
