# Peer review of "Reliable Method to Detect Alloy Soldering Fractures under Accelerated Life Test"

_applsci, doi:10.3390/app9163208_

Round 1

Reviewer 1 Report

In this paper authors presents the results of investigations on the development and design of the Accelerated Life Test (ALT) and its approach to the waste of material and its decrease. This test is important for these electronic works in which the components were built based on an LED and a diode.

I haven't significant remarks. However, it would be interesting to specify changes in alloy composition during soldering and exploitation in longer time, and their impact on the results of the proposed test.

In my opinion, the paper may be recommended for publication in the Electronics/Applied Sciences in the present form.

Author Response

In this paper authors presents the results of investigations on the development and design of the Accelerated Life Test (ALT) and its approach to the waste of material and its decrease. This test is important for these electronic works in which the components were built based on an LED and a diode. 

I haven't significant remarks. However, it would be interesting to specify changes in alloy composition during soldering and exploitation in a longer time, and their impact on the results of the proposed test.

In my opinion, the paper may be recommended for publication in the Electronics/Applied Sciences in the present form.

A: Thank you for your comments. In the new version of the paper we are specifying the changes in alloy composition during soldering and exploitation in a long time by commenting on the impact results of the test in the Discussion Section.

The main focus of the tests are to obtain data quickly, which, properly modeled and analyzed, provide desired information about the life of a product under normal conditions of use. This process was developed to demonstrate that it is possible to find minor faults in LED-diode, electronic cards and resistance. There were no changes in the chemical composition as evidenced by micrographs. There was mechanical wear: cracks, ruptures. Due to the thermal shock due to temperature and vibration, the porosity of the weld became larger due to the detachment of the weld. Other methods applied to need more tests.

Reviewer 2 Report

The article falls within the scope of the journal Applied Sciences, special Issue "Structural Integrity of Aluminium Alloys".

The manuscript is of sufficient scientific interest and has originality in its technical content. However, the manuscript requires minor corrections. Remarks and comments are presented below.

Specific comments:

In which place the gas constant R is used in formula (4)?

Line 167: k is constant or temperature dependent ?

On what basis the values Ea and k were assumed in formula (5)?

Chapter 7 should be divided into two separate chapters: Discussion of results and Conclusions. The authors should carefully analyze the text in Chapter 7 and clearly separate the discussion of the results from the conclusions. For example, the last paragraph of subchapter 7.1 are conclusions, and the first paragraph of subchapter 7.2 is a discussion of results in relation to the achievements presented in other works.

All abbreviations in the article should be explained in detail at the first use.

There are a lot of editorial (eg no spaces) and punctuation errors in the work. Authors should carefully check the whole text.

The manuscript should be checked by a native speaker.

Author Response

The article falls within the scope of the journal Applied Sciences, Special Issue "Structural Integrity of Aluminium Alloys".

The manuscript is of sufficient scientific interest and has originality in its technical content. However, the manuscript requires minor corrections. Remarks and comments are presented below.

Specific comments:

In which place the gas constant R is used in formula (4)?

A: Thank you for your comments. In the new version, the Equation 4. Was corrected, since R was a mistake in the first version 

    Line 167: k is constant or temperature dependent?

A: Thank you for your comments. In the new version of the paper, we are specifying that k is a constant, k = Boltzmann’s constant (physical constant that relates absolute temperature and energy). Please check the reference [12].

On what basis the values Ea and k were assumed in formula (5)?

A: Thank you for your comments. We are mentioning that Ea is the activation energy is known as the voltage threshold of diodes in general, is equal to 0.7 V.  The Arrhenius acceleration model and Ideal Gas Law, the Bolzmman´s constant (k) has a value of 8.6x10-5 = 0.0000086.

Chapter 7 should be divided into two separate chapters: Discussion of Results and Conclusions. The authors should carefully analyze the text in Chapter 7 and clearly separate the discussion of the results from the conclusions. For example, the last paragraph of subchapter 7.1 are conclusions, and the first paragraph of subchapter 7.2 is a discussion of results in relation to the achievements presented in other works.

A: Thank you for your comments. In the new version of the paper, we rearrange the Chapter 7 as the reviewer suggested.

All abbreviations in the article should be explained in detail at the first use.

A: Thank you for your comments. A glossary of terms has been added at the end of the paper.

There are a lot of editorial (eg no spaces) and punctuation errors in the work. Authors should carefully check the whole text.

A: Thank you for your comments. All those issues have been corrected.

The manuscript should be checked by a native speaker.

A: Thank you for your comments. A native speaker has reviewed the new version of our paper.